# Reduction of Postharvest Quality Loss and Microbiological Decay of Tomato “Chonto” (*Solanum lycopersicum* L.) Using Chitosan-*E* Essential Oil-Based Edible Coatings under Low-Temperature Storage

**DOI:** 10.3390/polym12081822

**Published:** 2020-08-13

**Authors:** Yeimmy Peralta-Ruiz, Carlos David Grande Tovar, Angie Sinning-Mangonez, Edgar A. Coronell, Marcos F. Marino, Clemencia Chaves-Lopez

**Affiliations:** 1Faculty of Bioscience and Technology for Food, Agriculture, and Environment, University of Teramo, 64100 Teramo, Italy; yyperaltaruiz@unite.it (Y.P.-R.); cchaveslopez@unite.it (C.C.-L.); 2Facultad de Ingeniería, Programa de Ingeniería Agroindustrial, Universidad del Atlántico, Puerto Colombia 081008, Colombia; alsinning@mail.uniatlantico.edu.co (A.S.-M.); edgar230710@hotmail.com (E.A.C.); mmarinojimenez@gmail.com (M.F.M.); 3Grupo de Investigación de Fotoquímica y Fotobiología, Universidad del Atlántico, Puerto Colombia 081008, Colombia

**Keywords:** antifungal, chitosan coatings, *Ruta graveolens* essential oil, postharvest quality, *Solanum lycopersicum*

## Abstract

The tomato (*Solanum lycopersicum* L.) is one of the many essential vegetables around the world due to its nutritive content and attractive flavor. However, its short shelf-life and postharvest losses affect its marketing. In this study, the effects of chitosan-*Ruta graveolens* (CS + RGEO) essential oil coatings on the postharvest quality of Tomato var. “chonto” stored at low temperature (4 °C) for 12 days are reported. The film-forming dispersions (FFD) were eco-friendly synthesized and presented low viscosities (between 0.126 and 0.029 Pa s), small particle sizes (between 1.29 and 1.56 μm), and low densities. The mature index (12.65% for uncoated fruits and 10.21% for F4 coated tomatoes), weight loss (29.8% for F1 and 16.7% for F5 coated tomatoes), and decay index (3.0 for uncoated and 1.0 for F5 coated tomatoes) were significantly different, indicating a preservative effect on the quality of the tomato. Moreover, aerobic mesophilic bacteria were significantly reduced (in five Log CFU/g compared to control) by using 15 μL/mL of RGEO. The coatings, including 10 and 15 μL/mL of RGEO, completely inhibited the mold and yeast growth on tomato surfaces without negatively affecting the consumer acceptation, as the sensorial analysis demonstrated. The results presented in this study show that CS + RGEO coatings are promising in the postharvest treatment of tomato var. “chonto”.

## 1. Introduction

The tomato (*Solanum lycopersicum* L.) is one of the many essential vegetables around the world, with a production of about 163 million tons per year and a high content of nutritious molecules including vitamin C, and E, β-carotene, lycopene, thiamin, riboflavin, and niacin, among others [1,2,3].

However, the high production brings quality issues, especially in the postharvest stage, where tomato decay is a significant challenge in most developing countries since it is a very high perishable crop as a result of its high moisture content [4,5]. Developing countries also find severe problems in the postharvest tomato. Up to 30% of the tomato harvested crop may be lost during postharvest handling, mainly due to microbiological deterioration caused by fungus-like *Rhizopus stolonifer*, *Alternaria alternata*, and *Botrytis cinerea* [6,7,8,9]. Some figures even account for 55% of losses of the total harvestable tomato per year, such as in the Australian market, for example [10].

Colombia is not an exception, where the tomato production accounts for a total cultivated area of 4500 hectares. However, 50% of the harvesting tomato is lost because of fungal decay in postharvest stages [11]. Fungicides are typically used to prevent fungal infection, like iprodione (Rovral), dichloran, fludioxonil, and fenhexamid, which eventually degrade into toxic compounds and generate pollution in the environment, complications in human health, and ultimately, resistant fungal strains [12,13]. Alternative strategies for fungal decay are proposed, like ozone (O_3_), modified atmosphere packaging (MAP), ultraviolet-C (UV-C) light, gamma irradiation, and bioactive natural compounds [14]. However, the uses of ultraviolet or gamma irradiation have grave concerns for human health, while ozone introduction is still costly.

A safer, cheaper, and environmentally friendly approach is found in the application of edible coatings to the surface of fruits. Usually, the layers are prepared from natural biopolymers such as polysaccharides and natural ingredients, taking advantage of their packaged structure based on a hydrogen-bonding network with an improved barrier to oxygen, moisture, and solute migration [14,15,16] which makes them attractive in fruit applications.

Chitosan has been previously used as an edible coating in several fruits based on its excellent antimicrobial and biocompatibility properties [16,17]. However, its hydrophilic nature forces the introduction of hydrophobic compounds such as some essential oils, which also provide antioxidant, antibacterial, and antifungal properties to the food during the postharvest stage [14,18,19,20]. Although several chitosan-essential oil strategies have been reported to control fungal decay of tomatoes, many of them were applied during preharvest stages, with severe complications in the growth of the tree leaves. On the other hand, few studies have been addressed during the postharvest stage directly [21,22,23,24,25,26]. However, some of them were applied to cherry tomatoes, and others did not show complete fungal inhibition. For example, studies using coatings of chitosan–essential oils of (lemongrass) or *Thyme* essential oil in combination with propolis were reported efficient in delaying the growth of *R. stolonifer* and preserving the quality of fresh tomato (*Lycopersicon esculentum* Mill.) fruit at room temperature (25 °C) storage [25,26]. In the same way, chitosan combination with starch demonstrated an excellent effect in weight loss and firmness conservation without microbial infection at room temperature [27].

In recent years, chitosan-based nanoemulsions have emerged as an alternative to the conventional biofilms, presenting some advantages such as the allowance of a higher transfer area and higher reaction rates, a higher solubility, improved bioavailability, optical transparency [28]. Moreover, they can limit the non-essential reactions with other components in the case of the food applications, as well as inhibit degradation during and after consumption [29]. Different studies present the chitosan-essential oils based nanoemulsions as an alternative to avoid the decay of fruits, with the critical advantage not to generate changes in the organoleptic conditions of the foods where they are applied [28].

Some studies have reported the effect of chitosan-based nanoemulsions incorporated with nutmeg seed essential oils and Zatariamuti flora essential oil in strawberries, with thyme essential oil in avocadoes, and with lemongrass essential oil in grape berries [30,31,32,33]. In general, the emulsions presented good antimicrobial activity and physicochemical property-preservation such as color, firmness, total soluble solids, and weight in the fruits where they were applied. Regarding tomatoes, Robledo et al. [34] reported a decrease in the Botrytis cinerea growth in cherry tomatoes with the use of chitosan–thymol essential oil-based nanoemulsion as the coating. Despite all the information published, the study of the effects in the mold and microbial spoilage in postharvest stage on tomato var. “chonto,” as well as the impact on the postharvest quality, of chitosan-*Ruta graveolens* essential oil (RGEO) coatings have not been reported yet.

The proposed study represents an excellent option to complement the antimicrobial activity of chitosan and extend the postharvest stability of tomato “chonto” under refrigeration conditions, improving the stability of tomatoes during 12 days of storage. Based on the vigorous antifungal activity of some RGEO components, the efficiency of CS + RGEO to increase the stability of fruits has been demonstrated by our group in guavas to control *Colletotrichum gloesporioides* fungi growth and quality aspects [16], cape gooseberries for microbial and quality assessment, and papayas [35,36,37]. This is the first time that the application of CS + RGEO coatings is reported in Tomato var. “chonto” (*Solanum lycopersicum* L.) to evaluate the effect in quality aspects and as a postharvest strategy. The study could be beneficial for farmers and producers in Colombia and developing countries, promoting their exportation capacity around the world.

## 2. Materials and Methods

### 2.1. Fruit Samples

Two hundred tomatoes “Chonto” (*Solanum lycopersicum* L.), which were healthy fruits (absence of peel damage and fungal infection) with visual uniformity in color and size, were selected in a maturation stage of four according to the USDA standard tomato, for color classification from a local market of Soledad, Atlántico, Colombia, and conducted to the laboratory in less than one hour [38]. The tomatoes were washed with a 100 mg/L solution of sodium hypochlorite, air-dried at ambient conditions, and stored at a temperature of 4 ± 0.2 °C until use [39].

### 2.2. Preparation of Edible Coatings

The preparation of the film-forming emulsions (FFE) based on CS + RGEO followed our previous reported methodology [37] and as shown in Figure 1, mixing a specific amount of chitosan (degree of deacetylation = 85%, Mw = 190,000–310,000 Da, Sigma-Aldrich, Palo Alto, CA, USA), with a certain volume of 0.1 M acetic acid a to obtain a 2% *w/v* solution. Addition of 2.5% *v/v* of glycerol and Tween 80 (1% *v/v*) concerning the final volume of RGEO (Krauter, Bogotá, Colombia) was followed to the complete homogenization using an IKA T25-Digital Ultraturrax (IKA, Staufen, Germany) at 7000 rpm for two minutes. RGEO previously characterized, was added to reach final concentrations of 0.5, 1.0, and 1.5% *v/v*, with respect to the chitosan solution [16]. Air bubbles were removed by freeze—thaw cycles from the emulsions.

### 2.3. Application of Edible Coatings to Tomatoes

Tomatoes were coated by dip coating in the different emulsions in a cell design for this proposal and detailed in Section 2.3.1. The different formulations consisted of one control consisting in a tomato dipped in pure distilled water (F1) and four different formulations (F2 = CS, F3 = CS + RGEO 0.5%, F4 = CS + RGEO 1.0%, F5 = CS + RGEO 1.5%), cells of glass with dimensions of 15 cm × 9 cm × 12 cm during five minutes. Afterward, the tomatoes were air-dried for 60 min and stored in boxes of polyethylene terephthalate (PET) under refrigerated conditions (4 ± 0.2 °C) for 12 days. Evaluations of the physical–chemical properties of tomatoes occurred at days 0, 3, 6, 9 and 12.

#### 2.3.1. Design Immersion Cell

Dip coating has been extensively utilized in fruits being an advantageous and facile method that does not need sophisticated equipment, and it is much more convenient and effortless than other approaches [40]. However, to control critical parameters as designated time, homogenous deposition, and drainage time, a cell was designed as seen in Figure 2. The cell consists of a rectangle glass bucket with a capacity of 1000 mL, a stainless-steel metal basket to control the drainage of residual emulsion and, a lower tray for fruit support and subsequent drying. Immersion and drainage time were carried during three and two minutes, respectively.

### 2.4. Physical–Chemical Properties of the FFE

The particle size, density, viscosity, and non-volatile compounds of the emulsions were determined according to the methodologies reported previously [16]. Briefly, the particle size was measured with a laser diffractometer was used (AIMSIZER 2011, Dandong, Liaoning, China). The apparent viscosity with a Brookfield LVF (Toronto, ON, Canada) viscometer. The density determination followed the ISO 8655-2 [41], according to Equation (1).
(1)$$d=\frac{P_{1}}{V_{1}}$$
where *P*_l_ is the weight of the sample (g) and *V*_l_ the volume of sample (1 mL) at 25 °C (g/mL).

The non-volatile content (S%) of the FFE were calculated from Equation (2):*S* (%) = ((*P_s_* − *P_d_*)/(*P_m_* − *P_d_*)) × 100(2)
where *P_d_*_,_ is the weight of the aluminum disk (g), *P_m_* is the weight of the sample and aluminum disk (g), and *P_s_* is the weight of the dried sample and aluminum disk (g).

### 2.5. Postharvest Quality Properties of Tomatoes

#### 2.5.1. pH and Soluble Solids (SS)

The pH was determined with a potentiometer Thermo Fisher Scientific Orion (Waltham, MA, USA) calibrated with 4, 7, and 10 calibration kit Thermo Fisher Scientific Orion (Waltham, MA, USA). Samples used consisted of ten grams of homogenized fruit in 100 mL of distilled water [37]. Determination of total soluble solids (TSS) (%) used a BRIXCO 0–90% brand refractometer.

#### 2.5.2. Titratable Acidity (TA)

The acidity (citric acid %) was determined using potentiometric titration with 0.1 N NaOH, five grams of fruit homogenized, and 50 mL of distilled water, according to Equation (3) [37]:(3)$$Citric\;acid\;\left(\%\right)=\left(\frac{V_{1}-N}{V_{2}}\right)\times K\times 100$$
where, *V*_1_ is the volume of NaOH used (mL), *V*_2_ is the volume of sample (mL), *K* is the equivalent weight of citric acid (0.064 g/meq), and *N* is the normality of NaOH (0.1 meq/mL).

#### 2.5.3. Maturity Index (*MI*)

It was calculated using Equation (4) [16]:(4)$$MI=\frac{\%Brix}{\%Acid}$$
where, % Brix is the total soluble solids (%) measured as degree Brix determined as is showed in the Section 2.5.1 and %Acid is the titratable acid measured as citric acid % calculated in the Section 2.5.2.

#### 2.5.4. Weight Loss Percentage

Fruits were weighed at day 0 and days 3, 6, 9, and 12 of storage. The difference between initial and final fruit weight was considered as total weight loss during that storage interval and expressed as a percentage of weight loss on a fresh weight basis [42].

#### 2.5.5. Color Analysis

The color CIELab system coordinates *a* * (−green, +red) and, *b* *(−blue, +yellow), and the red–yellow ratio (*a* */*b* *) were determined through a calibrated computer vision system using the professional photo editing software Adobe Photoshop^®^ CS5 test version (Adobe System Inc. 2015 Edition, San Jose, CA, USA). Quantification of the coordinates *a* * (−green, +red) and, *b* *(−blue, +yellow) were performed at several points across the equatorial and apical zone of the fruit. The *a* * and *b* * were obtained directly, and then the red-yellow ratio (*a*/*b*) was reported to indicate the redness of tomatoes [43]. The color variation in the samples was analyzed accordingly to [44], identifying the color parameters in the tomatoes according to their maturation stage. The total color difference (Δ*E**) was calculated using the following Equation (5) [37]:(5)$$∆E^{*}=\sqrt{{\left(∆L^{*}\right)}^{2}+{\left(∆a^{*}\right)}^{2}+{\left(∆b^{*}\right)}^{2}}$$

#### 2.5.6. Firmness Analysis

The puncture firmness of all fruit samples was measured with a fruit pressure tester Facchini model FT 011 (0–11 pounds) (Facchini, Alfonsine, Italy) with a plunger with a diameter of 8 mm inserted into the fruit skin manually. The penetration force readings were in kilogram-force (kg f) and converted to Newton (N) units. The firmness was tested on three sides of each fruit.

### 2.6. Antimicrobial Assay

The microorganism count was performed by two replicates at days 0, 3, 6, 9 and 12 using a reported methodology [45]. For the mold and yeast count, Potato-glucose agar (PDA) as media with 10% tartaric acid (*w/v*) were used, and the plates were incubated at a temperature of 25 ± 0.2 °C for five days. Total viable count (CFU/mL) of mesophylls was performed using a Plate Count Agar (Difco, Kansas City, MO, USA) after incubation at 25 °C for 48 h.

#### 2.6.1. Decay Index (*DI*)

The *DI* was evaluated visually according to the methodology proposed by Perdones et al. [46] following the scale of Figure 3. The results of fungal presence, mechanical damage, and physical skin deterioration were calculated using Equation (6):(6)$$Decay\;Index=\frac{1n+2n+3n+4n}{N}$$
where *n* = number of fruits classified in each level of the damage scale and *N* = number of total fruits analyzed in each treatment per day. The decay index of fruits was evaluated on days 0, 3, 6, 9, and 12.

#### 2.6.2. Disease Damage Incidence

The incidence of fungal diseases was measured following the method reported by Badawy and Rabea [47], with some modifications. The percentage of incidence was calculated using Equation (7):(7)$$Disease\;damage\;incidence\;\left(\%\right)=\frac{number\;of\;infected\;tomatoes}{total\;samples\;per\;treatment}\times 10$$

### 2.7. Sensorial Activity

The test was carried out considering the standard ISO 11035:1994 [48]. The experiments were performed in the nutrition and dietetics laboratory of the Universidad del Atlántico, with 50 people non-trained assessors during days 0, 5, and 10 of storage. The samples were cut in slides with the same size. The peel was not removed by the tomato-consumers’ preferences in Colombia. The assessors were informed of the test methodology, and additionally signed an informed consent that contained data on the reagents used in the preparation of the emulsions and the risks of exposition. In the test, the attributes of pulp color, flavor, aroma, texture, and gloss were evaluated. Panelists were asked to score the difference between samples where 0–2 represented extreme dislike; 3–5 fair; 6–8 good; and nine excellent for each attribute.

### 2.8. Statistical Analysis

The analysis of variance (ANOVA) and the least significant difference (LSD) method of multiple comparisons, with a 95% confidence level (α = 0.05), were used to assess the effect of edible coatings on the response variables described previously. The Stat graphics Centurion XVI program was used for these statistical analyzes.

## 3. Results

### 3.1. Characterization of the Film Forming Emulsions (FFE)

Data regarding the characterization by mass spectrometry-gas chromatography (MS-GC) of the RGEO were reported elsewhere [16]. The temperature and pH usually are adjusted to facilitate chitosan solubility. For that reason, the pH of the emulsions was acidic in all cases and is similar without significant differences (Table 1).

The density values of the FFE demonstrated significant reductions (*p* < 0.05) with the introduction of the RGEO, as previously observed for other authors [49]. Similarly, the incorporation of RGEO had significantly reduced (*p* < 0.05) the apparent viscosity of the FFE compared to F1 (0% of RGEO). The lower value was obtained in the emulsion with the highest concentration of RGEO (28.7 ± 0.2 cP) and the higher value in the chitosan solution (106.0 ± 0.1 cP).

On the other hand, CS + RGEO emulsions showed a significant increase (*p* < 0.05) in the total solid percentages due to superior retention of the oil in the polymer matrix until 1.0% of RGEO. After that percentage of RGEO, no significant increase (*p* < 0.05) of the total solid amount was observed.

### 3.2. Physical–Chemical Analysis of Fruits

The physicochemical properties determine the quality, shelf life, sensory aspects, and the post-harvest handling of the fruits and are very important when evaluating the effectiveness of the coatings. The pH, titratable acidity, soluble solids, and mature index of the tomato fruits were determined at days 0, 3, 6, 9, and 12 of storage at a cold temperature (4 °C) (Table 2).

#### 3.2.1. pH Analysis

The pH usually increases during the maturation of a climacteric fruit due to the organic acids’ consumption for the metabolic processes during fruit respiration. However, the pH did not change for all the days and formulations until day 12 (Table 2), where F4 and F5 showed a significantly (*p* < 0.05) lower pH than the other formulations. Moreover, all the formulations presented significant differences concerning the other days.

#### 3.2.2. Soluble Solids (SS)

Sugars are the main components of the SS in tomato fruits [50]. Soluble solids (SS) are more often measured than total soluble solids, despite both indexes being correlated. SS analysis demonstrated no differences between treatments on the same day except on day 3 for F2 and F5. In the present study, SS increased until day 12 of storage at a cold temperature (Table 2), as a consequence of hydrolysis of carbohydrates during the ripening process [20].

#### 3.2.3. Titratable Acidity (TA)

TA is related to the organic acid contents in tomato fruits, mainly citric acid and ascorbic acid [26,35]. TA decreased in all coated tomatoes during the storage at low temperatures (Table 2). However, there were no significant (*p* < 0.05) differences between the treatments on the same day, which is related to the pH and SS behavior.

#### 3.2.4. Mature Index (*MI*)

Mature index (*MI*) (Table 2) showed significant differences (*p* < 0.05) between uncoated and coated samples after the third day of storage under low temperature, which could be related to a barrier effect produced by the coatings decreasing the metabolism of the fruits [51]. During days 6, 9, and 12, there were also significant differences between the coated tomatoes (F4 and F2, F3, and F5). There were also lower *MI* in day 6 for F3, and days 9 and 12 for F4.

#### 3.2.5. Weight Loss Percentage

Weight loss percentages in climacteric fruits are usually related to water losses caused by the water vapor exchange (transpiration) and respiration of the fruits [24]. From Table 3, there were significant differences (*p* < 0.05) in weight losses between coated and uncoated tomatoes in days 3, 6, 9, and 12 of storage at low temperatures. On day 12, F1, F2, and F4 presented significant differences (*p* < 0.05).

#### 3.2.6. Color Parameters Analysis

The analysis of the color in fruits impacts the marketing of the fruits. A consumer might be attracted or reject some fruit only by visual examination. We determined the color changes using the CIE L **a* * *b* * scale, where *a* * and *b* * are the chromaticity coordinates (rectangular coordinates), and L * the lightness. On the other hand, +*a* * is the red direction, −*a* * is the green direction, +*b* * is the yellow direction, and −*b* * is the blue direction, and lightness value L * represents the darkest black at L * = 0 and the brightest white at L * = 100. Table 4 shows a significant increase (*p* < 0.05) for the red/green coordinates (*a* *) and the yellow/blue coordinates (*b* *) until day 6 of the storage at low temperatures for all treatments. However, decreased values for the coordinate degradation of the molecules is responsible for the red color in tomatoes. Concerning the coordinate L *, a reduction was observed with the storage time, and it only presented significant differences among treatments F1 and F5 with F2, F3, and F4 for day 9. At the end of the storage time on day 12, the procedures with the lower values for coordinates *a* *, *b* *, and L * were F3, F4, and F5, indicating a slower ripening process. The control fruit and chitosan-coated fruit showed high Δ*E* values, whereas in treated fruits, this parameter was less affected. However, the best red/yellow ratios were exhibited by F2 and F4 and the minor Δ*E* by F3, F4, and F5 meaning that CS + RGEO did not negatively affect the color of tomatoes.

#### 3.2.7. Firmness Analysis

During the storage time of tomatoes at low temperatures, a significant decrease (*p* < 0.05) in hardness was observed, as shown in Table 5 for all the treatments. However, on day 12 of the storage time, the uncoated and coated tomatoes with F4 and F5 exhibited significant differences (*p* < 0.05) in the firmness values that correlate well with the lower weight loss of these samples by an improved barrier effect caused by the increased RGEO content.

### 3.3. Antimicrobial Assay

In the present study, the ability of CS + RGEO coatings to delay or inhibit spoilage microorganisms on tomato var. “chonto” was assessed. Aerobe mesophilic bacteria count describes the population of bacterial colonies typically corresponding to coccus, bacillus, and spiral bacteria that it is present in high levels will indicate poor hygienic conditions [52]. The effect of the application of CS + RGEO coatings on tomatoes is observed in Table 6. The statistical analysis showed that there are significant differences (*p* < 0.05) between treatments on the same day of storage and between days at a 95% confidence level. On the first day of storage, there was no growth for aerobic mesophylls with F4 and F5, while F3 decreased the growth by almost 2.0 Log CFU. From day 3 to day 12, all the treatments were unable to inhibit the bacterial growth entirely, but F5 was able to keep the population lower than two Log CFU. In comparison, F4 kept the population under 3.0 Log CFU, demonstrating a robust antibacterial effect of RGEO when it is used at 10 (1.0%) and 15 μL/mL (1.5%) for F4 and F5, respectively. The uncoated fruits and F2 (only chitosan) account for an extremely high population of aerobic mesophylls (7.0 and 6.7 Log CFU/g, respectively).

The results of the growth of spoilage fungi on tomatoes surfaces are observed in Table 6. Significant reduction (*p* < 0.05) in the mold population occurred for F4 and F5 (10 and 15 μL/mL of RGEO). Complete inhibition growth of fungi on day 12 of storage was observed. On the other hand, CS alone (F2) could not inhibit fungi growth (5.3 Log CFU/g) while the control (F1) had 5.6 Log CFU/g. Neither way, F3 (5 μL/mL of RGEO) only reduced to 3.8 Log CFU/g. Those results indicate a minimum inhibitory concentration for RGEO around 5 to 10 μL/mL.

#### 3.3.1. Decay Index (*DI*)

*DI* only exhibited significant differences after the sixth day between uncoated and coated fruits (Figure 4). However, non-significant differences were observed between F2, F3, F4, and F5 on days 6 and 9. At the end of the storage time (Day 12), there were significant differences (*p* < 0.05) between F1 and F2 with F3, F4, and F5, indicating a beneficial effect with the incorporation of RGEO to the CS coatings.

#### 3.3.2. Disease Damage Incidence

As shown in Figure 5, on the twelfth day, disease damage incidence was 100% in uncoated tomatoes, while treatments with CS + RGEO 0.5% reduced the disease incidence for about 80%. On the contrary, CS+RGEO 1.0 and 1.5% inhibited the disease damage incidence totally in Tomatoes. Significant differences (*p* < 0.05) were observed between F1 and F2 with F3, F4, and F5, indicating a protective effect of the CS + RGEO against fungal tomato diseases.

### 3.4. Sensorial Analysis

A sensorial analysis of the coated and uncoated tomatoes was assessed on days 0, 5, and 10, to verify that no adverse effects on the quality and acceptability of the tomatoes occurred. Sensory aspects may include appearance, color, flavor, and texture, which probably remains the most required attribute strongly affecting consumer decision to purchase the product [53]. The results of the sensory analysis are represented in a hedonic curve, as seen in Figure 6. The treatments that showed lower effects on sensory characteristics were F1 and F2. In contrast, treatments F3, F4, and F5 were negatively affected by the flavor attribute. However, in the texture, aroma, and gloss attributes, there were no significant differences during the treatments.

## 4. Discussion

The present study used chitosan of medium molecular weight, taking advantage of excellent film-forming properties, superior mechanical characteristics, improved gas barrier, lesser flavor and aroma loss, and higher humidity resistance capacity than chitosan of low molecular weight [35]. On the other hand, despite the controversy that chitosan of low molecular weight presents better antimicrobial activity due to electrostatic interactions with cell membranes of the microorganisms [54], the activity of chitosan-medium molecular weight has also shown excellent antimicrobial activity due to adsorption on the cell surface for Gram-positive bacteria and fungi [55]. Therefore, to maximize the antimicrobial and barrier properties of chitosan, as well as its biocompatibility, it was combined with the high antifungal and hydrophobic power derived from the terpenoid and ketone-type components of RGEO [16,37,56,57,58,59].

Preparation of stable and useful coatings usually is achieved using materials that are easily dissolved in water, while some additives are emulsified (like plasticizers and stabilizing agents) using surfactants, which in turn decrease the fruit ripening [60]. In this study, the emulsions presented excellent stability without any separation phenomena when they were observed after six months. Regarding the viscosity of the emulsions, chitosan at acidic pH has a cationic structure with a high viscosity, usually obtained for medium molecular weight chitosan. However, with the introduction of the RGEO, unexpectedly, the viscosity decreases. Similar results have been collected for other studies [49]. At the pH of chitosan solutions, several electrostatic interactions between chitosan chains and the main components of the essential oil occur, decreasing the net electric charge of the solution and leading to bigger droplet sizes, as reflected by the particle size measurements [49,61]. The chitosan interfacial adsorption on the oil droplets leads to stabilization of the emulsion [61,62].

Particle sizes increased with the RGEO content since chitosan chains are adsorbed on the oil droplet surfaces, including more and more oil droplets in a bridge mode until no more chitosan chains are available, leading to some flocculation of the oil droplets [63]. Rheology studies observed bimodal distributions of the ζ-potential because of the chitosan adsorption on the surface of some oil droplets. Regardless, some oil droplets without any chitosan adsorbed [62]. The particle charge affects the rheology of the emulsions by electroviscous effects, for instance, altering the viscosity and the droplet sizes [64]. CS + RGEO 1.5% does not present a total solid percentage increase in comparison with the other emulsions. This behavior is probably related to the oil evaporation (lower chitosan adsorbed on the oil interfaces lead to oil evaporation in the analysis). Moreover, oil droplets are adsorbed in the hydrophobic region of the chitosan through van der Waals interactions and hydrogen bonds between hydroxides and amines of the CS and ketones present in the oil.

The effects of CS + RGEO coatings on the physicochemical properties of tomato fruits were evaluated. A lower consumption of organic acids related to a lower pH for tomatoes coated with F4 and F5 at the end of the storage at cold temperature was observed. Other authors reported similar trends with chitosan-based coatings with pH also ranging between 4.0 and 4.6 [65,66]. Changes in the internal atmosphere could be the cause of the differences in pH, generally showing some correspondence with the titratable acidity. Another factor of the discrepancy of F4 and F5 on day 12 could be intrinsic variations in the composition of the evaluated fruits, which depends on edaphic–climatic (environmental) and fertilization of the fruits (cultural) aspects [66]. Similarly, the differences between F2 and F5 on the 3rd day are related to intrinsic variations instead of treatments themselves.

In the present study, despite that no clear trends were observed for SS during the experiments, no adverse effect in the SS was observed. Some authors have attributed the variations of the SS to changes on the electrical conductivity of soils derived from fertilization processes [66,67,68,69] or due to water flow restrictions derived from osmotic pressure effects of the high electrical conductivity [70].

On the other hand, Barreto et al. [24] indicated, for cherry tomatoes, the absence of a total soluble solid decreasing with tomatoes coated with chitosan-*Origanum vulgare* essential oil as compared to the uncoated fruits. They argued a reduction in fruit metabolism effect from the glucose and fructose levels measured. Different trends were obtained by other authors, where a decrease then follows an initial increase in the SS values [25,71].

A decrease of TA in the different treatments was observed, which usually occurred in the fruit ripening after organic acid consumption for the synthesis of sugars during the metabolic pathways [24,50]. The trend was not clear, but the reduction of the TA was lower for CS+RGEO-coated samples, indicating that the coated tomatoes (Table 2) suffered a slowdown of the metabolism by the barrier effect of the CS + RGEO coatings against oxygen, inhibiting the oxidation of the organic acids like ascorbic acid [51]. Similar results have been reported for other chitosan-essential oil systems in tomatoes and cherry tomatoes [65,66]. Usually, the decomposition of the organic acids (citric, pyruvic, lactic, among others) is used as a substrate for metabolic biochemical reactions, for ATP synthesis, or even in enzymatic reactions [24,70,72]. More moderate SS content and a higher TA are consistent with a reduction of the metabolism of the organic acids or intrinsic differences of the experimental units, as stated above. However, the coating of the fruits delays the ripening process similar to other studies, which is a beneficial result to control the postharvest decay of fruits [25,71]. The values obtained for MI could result in a higher acceptance of the consumer since a low level of titratable acidity and high content of soluble solids produces a better taste and aroma of tomatoes [66].

The decay index and disease damage incidence are usually due to weight loss, but in some cases, fungal colonization is observed, which also deteriorates the quality of tomatoes. From the results of the *DI* measurements, CS + RGEO 0.5% (F3) could be enough to delay the decay index and the incidence of the fungal infection. It is well known that chitosan-based coating reduces free radical presence, increases the disease resistance, and for its elicitor activity, induces the production of defense-related enzymes in fruits [73,74]. Moreover, essential oil addition, such as cinnamon and clove oils to the chitosan coatings, improves the antioxidant capacity of the fruits by inducing defense mechanisms to the fruits [60]. In a previous study, the preservation of the antioxidant capacity of cape gooseberries using CS + RGEO coatings was demonstrated, which also had a positive influence on the deterioration index of the fruits [35]. The preservative effect of CS + RGEO coating might be due to a free-radical scavenging ability of the essential oil [35]. Additionally, some authors reported an increased stimulated activity of defense enzymes like superoxide dismutase (SOD), catalase (CAT), and peroxide dismutase (POD) in plants by the application of essentials oils [75]. This could account for the lower decay index of the fruits, which is regulated by the concentration of reactive oxygen species (ROS) [76]. Finally, an increased antifungal activity due to the ketone components of RGEO affecting cell membranes contributes to the preservation [37].

The weight is a parameter crucial for consumer acceptance and could be directly related to the decay of the fruit quality and fungal infections [16]. The decrease in weight loss percentage in F3, F4, and F5 compared with the control is indicative of coatings efficiency for delay the gas exchanges due to a semi-permeable barrier effect that is reinforced against water by the hydrophobic character of the RGEO [24,46,77]. Usually, changes in fruit weight are related primarily to water loss since the loss of volatile molecules responsible for aroma and flavor is practically undetectable in weight [78]. On the other hand, firmness loss, which is correlated with the softening of the fruit, is considered one of the most important characteristics during fruit ripening [79]. In this regard, it is a fact that fungi take advantage of colonization of the fruit by delivering cell wall degrading enzymes (such as polygalacturonase, pectin methylesterase, and β-galactosidase) during colonization and infection [80,81,82]. Usually, chitosan-essential oil-based coatings reduce transpiration, providing turgor to the fruit cells, maintaining firmness [16]. In this study, in treatments F4 and F5, when fungal growth was not detected, less firmness loss was observed. This behavior could be due to some components of the essential oil with the ability to oxidize the fungi enzymes related to the fungal decay of fruit [16,37].

Additionally, fruits coated with CS + RGEO presented less color change, with an increase in the *b* *, and *a* * coordinates. The increase could be associated with the fact that lycopene (related to the red color) and β-carotene (compared with the orange color) achieve their concentrations peaks in the full ripening [83]. With red color increasing in tomatoes, a decrease in the L * value was also observed, indicating the darkening of the red color. The intensification generally occurs during the ripening of the tomatoes, as is shown with the results of Δ*E*. It is evident from the matrix of color differences between F1 with F3, F4, F5 that the color change above 5.39 to the control can be perceived by consumers and is associated with a higher ripening stage than coated tomatoes [76]. The chromophore degradation molecules like lycopene could be the main reason for the loss in color attributes, which could be delayed by coatings [27,33]. Metabolic reactions allow the color of the fruit to increase its intensity after chlorophyll degradation and lycopene synthesis [30]. From the results of Δ*E*, it is evident that coating has a beneficial effect on the reduction of color changes in tomatoes. A color change above 5.39, which was found in the control batch, can be perceived by consumers and could be associated with a higher ripening stage of the tomatoes [84]. The chromophore degradation molecules like lycopene could be the main reason for the loss in color attributes, which could be delayed by coatings [25,37]. Metabolic reactions allow the color of the fruit to increase its intensity after chlorophyll degradation and lycopene synthesis [35].

The quality and shelf life of tomatoes and climacteric fruits usually are reduced due to their high vulnerability to spoilage microorganisms such as bacteria, molds, and yeast [74]. Microbial spoilage in tomatoes is due primarily to fungal attacks of *Rhizopus stolonifera*, *Aspergillus niger*, *Penicillium expansum*, and *Botrytis cinerea* causing soft-rot, black, blue/green, and grey rot mold, respectively [85]. When thin chitosan-essential oil coatings, including antimicrobial agents, are applied on the surface of the fruits, they are very active in inhibiting spoilage microbial growth, especially against fungal colonization [14,37,55,86,87]. From the gas chromatography-mass spectrometry (CG-MS) analysis of RGEO [16], the relative amount demonstrated that the predominant amounts in the essential oil were: 2-undecanone (42.6%), 2-nonanone (23.5%), 2-decanone (4%), 2-nonanol (3%), 2-dodecanone (2.9%), and 2-tridecanone (2.5%) as the main components, accounting for the 78.5% of the essential oil. All those components are oxygenated terpenes, with five ketones and one alcohol. Strong antibacterial and antifungal activity has been previously reported for the two main parts (2-undecanone and 2-nonanone) of the *R. graveolens* essential oil [88]. Despite this, some controversy remains on the chitosan antimicrobial effects depending on the source, molecular weight, and deacetylation degree [14,74]. In the present work, chitosan did not show antibacterial activity. The strong effect in the current work was directly dependent on the RGEO content. Usually, the primary mechanism considered for antimicrobial activity of chitosan depends on the electrostatic interaction between the positively-charged amino groups of chitosan and the negatively-charged carboxylate groups of bacterial cell membranes, disrupting the cell [74]. However, the previous effect strongly depends upon the cell membrane composition. In our case, only the diffusion of the RGEO components inside the bacteria cell caused cell growth inhibition. This could be related to the capacity of the essential oil inhibiting enzymatic reactions of membrane synthesis. Moreover, essential oils also have the ability to affect permeability capacity of the membrane by bonding ergosterol, disrupting the microbial mitochondria by affecting enzyme mitochondrial bacteria and affecting the level of reactive oxygen species (ROS), which oxidizes protein, DNA, and lipids inside the cell [14].

In this study, we found that the antifungal activity of CS+RGEO is strongly dependent on RGEO content. The antifungal mechanism of essential oils might be related to the diffusion inside the cells, affecting cell membrane synthesis, mitochondrial function, DNA destruction, cell lysis, or inhibiting the sporulation and germination of spoilage fungi [14,89]. Previous studies on tomatoes and cherry tomatoes account for suitable antifungal activities of different chitosan–essential oil treatments [22,23,24]. Still, our present work demonstrates that low-temperature treatment, in combination with CS + RGEO treatments with a minimum of 1.0% of RGEO, were able to inhibit fungi growth on tomato fruits in situ completely.

The results of the hedonic evaluations showed low scores in flavor for F3, F4, and F5. This could be influenced by bitter herbal flavors probably caused by the essential oil presence. Higher organic acid content was created by the better barrier performance of the CS + RGEO-coated tomatoes, which simply could be removed by a washing procedure or peel removal. However, in the texture, regarding aroma and gloss attributes, there were no significant differences during the treatments, which indicates that the characteristics were preserved between the first and sixth days of storage and profiling the CS+RGEO coatings postharvest procedures for tomato var. “chonto.”

## 5. Conclusions

In the present work, we demonstrated a natural and eco-friendly synthesis of Chitosan-*Ruta graveolens* essential oil (CS + RGEO) coatings with small particle sizes, low viscosities, excellent stability, and easy application on fruit surfaces of tomato var. “chonto” for postharvest treatment. From the study of the physical–chemical characteristics of the fruits, it was evident that no adverse effect on the quality of the fruits was induced. On the contrary, the mature index, decay index, disease damage incidence, and color results correspond to less ripped fruits. In regards to weight loss, tomatoes coated with CS + RGEO 1.5% exhibited lower weight loss, demonstrating that coatings had a barrier effect, possibly due to a modified internal atmosphere with an improved hydrophobic characteristic that avoids the fruit decay.

Very interestingly, our results demonstrated an improved antimicrobial effect than other reports. Formulations that included 10 and 15 μL/mL of RGEO completely inhibit the mold and yeast microbial spoilage. Moreover, 15 μL/mL of RGEO decreased the aerobe mesophilic bacteria to lower than 2.0 log CFU/g, which corresponds to tomatoes available for human consumption.

The sensorial analysis of the coated fruits demonstrated that all the formulations were acceptable for the organoleptic attributes. Still, the flavor had lower acceptance at all days of investigation, possibly induced by the bitter taste of the essential oil and superior organic acid content. However, this might be simply improved by a washing process at the end of the storage time. The results presented in this study demonstrated that CS + RGEO coatings are promising in the postharvest treatment of tomato var. “chonto,” preserving the physical–chemical properties and delaying or inhibiting microbial spoilage growth without negatively affecting the fruit acceptation by consumers.

## Figures and Tables

**Figure 1 polymers-12-01822-f001:**
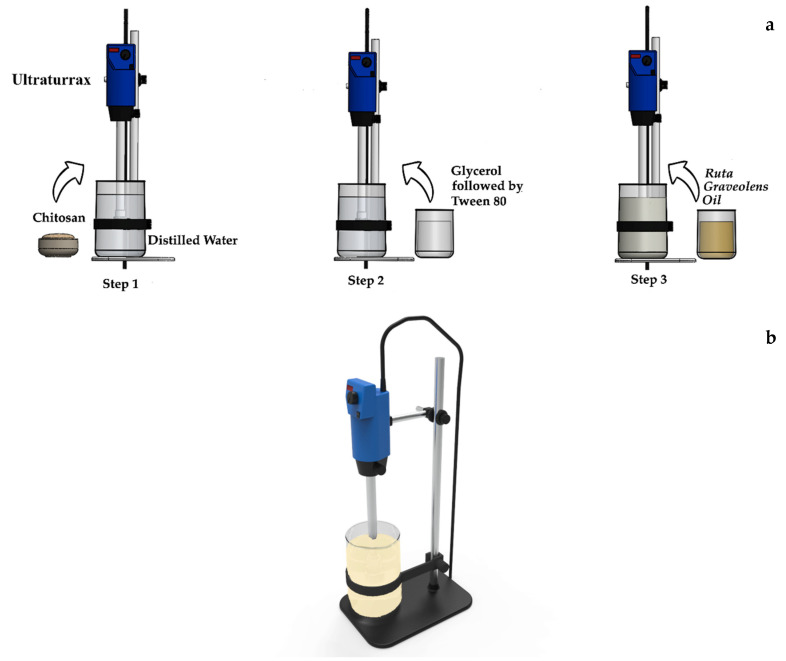
Preparation of film-forming emulsions, (**a**) A schematic view of the different steps emulsions preparation process, (**b**) a detailed picture of the emulsion preparation setup.

**Figure 2 polymers-12-01822-f002:**
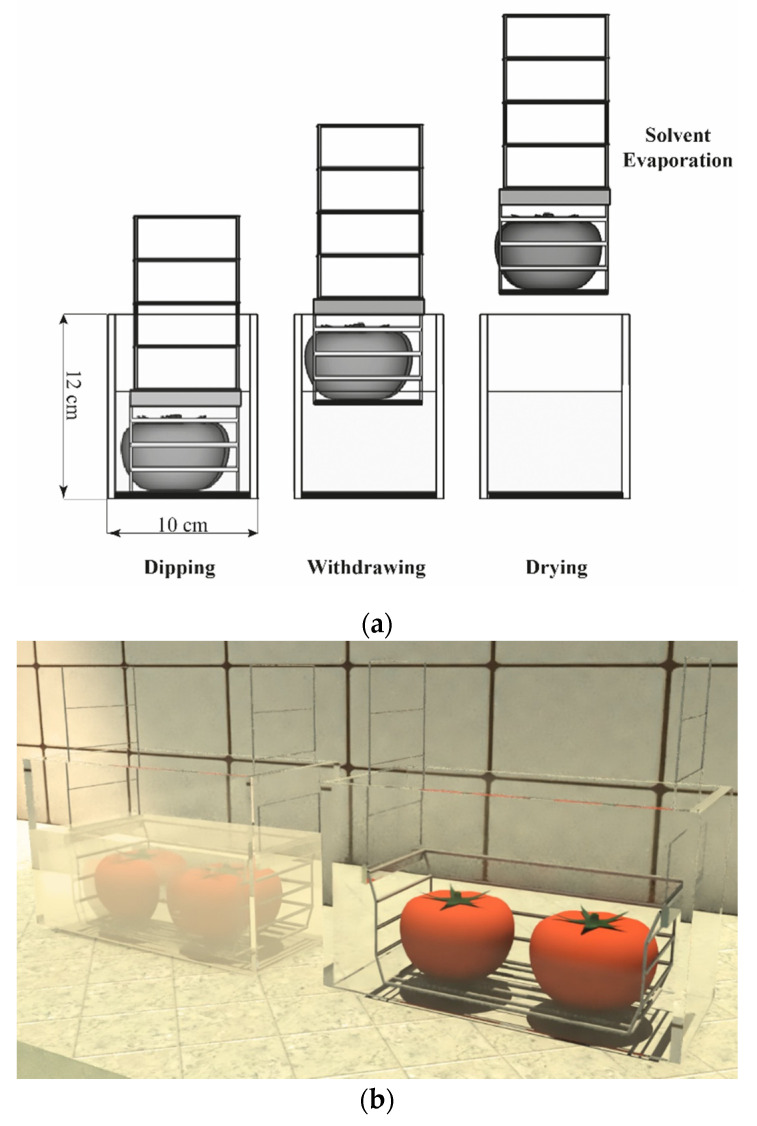
Design dip-coating cell, (**a**) A schematic view of the tomatoes dip-coating process, (**b**) a detailed picture of the cell.

**Figure 3 polymers-12-01822-f003:**
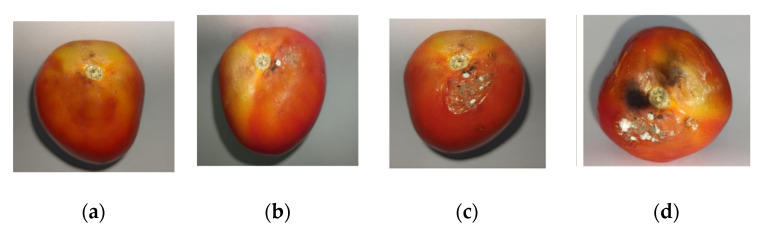
Hedonistic damage scale used for scale of the tomato, where (**a**) no damage (0% damage), (**b**) mild damage (10–15% damage), (**c**) moderate damage (25–50% damage), and (**d**) severe damage (>50% damage).

**Figure 4 polymers-12-01822-f004:**
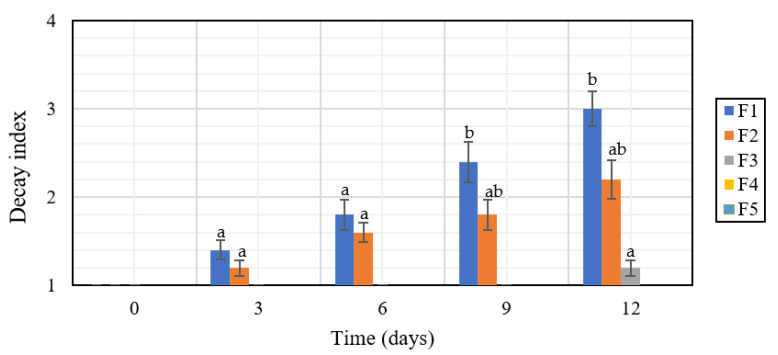
Decay index in *S. lycopersicum* with CS + RGEO coatings stored for 12 days at 4 ± 2 °C. F1 = Control; F2 = CS; F3 = CS + RGEO 0.5%; F4 = CS + RGEO 1.0%; F5 = CS + RGEO 1.5%. Mean values and intervals of LSD 95% according to the ANOVA test. The superscript letter (a–b) refers to the significant differences (*p* < 0.05) between each treatment.

**Figure 5 polymers-12-01822-f005:**
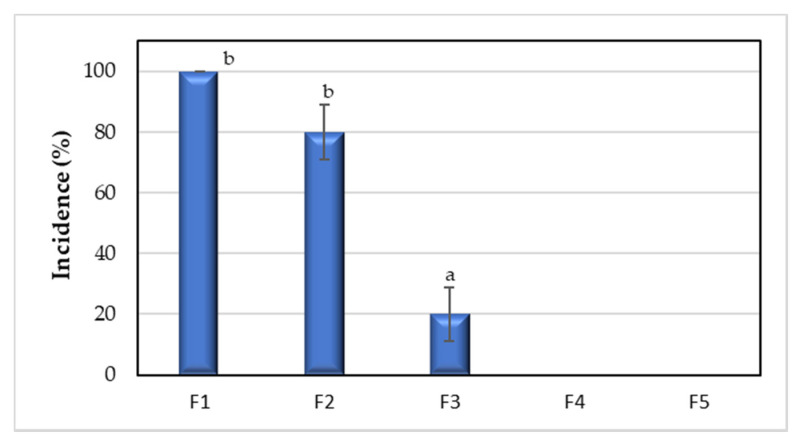
Disease damage incidence *S. lycopersicum* with CS + RGEO coatings stored for 12 days at 4.0 ± 2 °C. F1 = Control; F2 = CS; F3 = CS + RGEO 0.5%; F4 = CS + RGEO 1.0%; F5 = CS + RGEO 1.5%. Mean values and intervals of LSD 95% according to the ANOVA test. The superscript letter (a–b) refers to the significant differences (*p* < 0.05) between each treatment.

**Figure 6 polymers-12-01822-f006:**
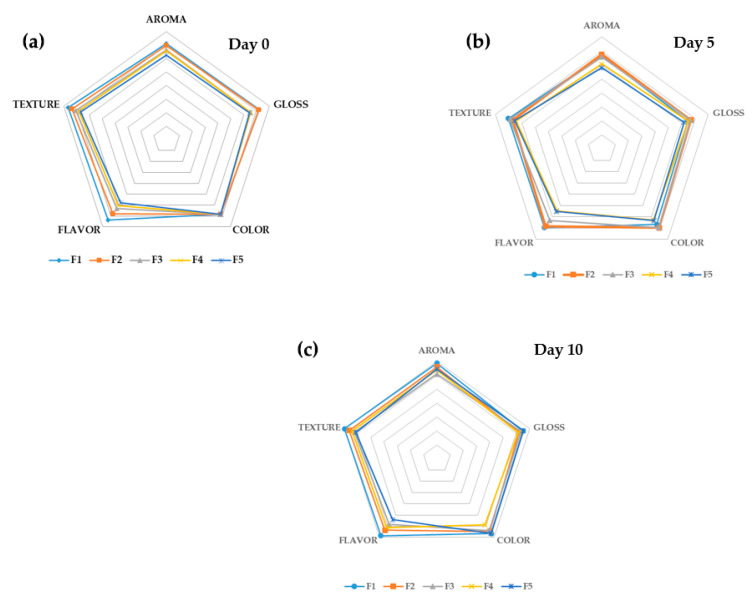
The hedonistic scale of the sensory analysis on days 0 (**a**), 5 (**b**), and 10 (**c**).

**Table 1 polymers-12-01822-t001:** Physical–chemical properties of the chitosan (CS) + *Ruta graveolens* L. essential oil (RGEO) FFE.

Essential Oil Content (%)	pH	ρ (kg/m^3^)	η_ap_ (100s^−1^) (Pa·s)	% Total Solids	Particle Size (μm)
0.0	4.42 ± 0.01 ^a^	1009.47 ± 0.17 ^d^	0.106 ± 0.001 ^d^	3.50 ± 0.02 ^a^	N.D.
0.5	4.44 ± 0.01 ^b^	1006.80 ± 0.005 ^c^	0.074 ± 0.005 ^c^	3.71 ± 0.01 ^b^	1.29 ± 0.25 ^a^
1.0	4.44 ± 0.01 ^b^	1002.43 ± 0.09 ^b^	0.066 ± 0.003 ^b^	3.87 ± 0.02 ^c^	1.43 ± 0.32 ^a^
1.5	4.45 ± 0.01 ^c^	1000.97 ± 0.102 ^a^	0.029 ± 0.019 ^a^	3.59 ± 0.02 ^a^	1.56 ± 0.12 ^a^

Note: The superscripts (a–d) in the same column refer to significant differences (*p* < 0.05) between each treatment.

**Table 2 polymers-12-01822-t002:** Physical–chemical characteristic in *S. lycopersicum* with CS + RGEO coatings stored for 12 days at 4 ± 2 °C. F1 = Control; F2 = CS; F3 = CS + RGEO 0.5%; F4 = CS + RGEO 1.0%; F5 = CS + RGEO 1.5%. Mean values and intervals of LSD 95% according to the analysis of variance (ANOVA) test. The superscript letter in the same column (a–b) refers to the significant differences (*p* < 0.05) between each treatment.

Day	Treatment	pH	TA (% Citric Acid)	SS (%)	Mature Index (%)
	F1	4.53 ± 0.07 ^a^	0.46 ± 0.01 ^a^	3.33 ± 0.29 ^a^	7.23 ± 0.01 ^a^
	F2	4.61 ± 0.06 ^a^	0.44 ± 0.01 ^a^	3.17 ± 0.29 ^a^	7.19 ± 0.39 ^a^
0	F3	4.70 ± 0.11 ^a^	0.39 ± 0.02 ^a^	3.00 ± 0.0029 ^a^	7.71 ± 0.76 ^a^
	F4	4.68 ± 0.09 ^a^	0.42 ± 0.003 ^a^	3.00 ± 0.0029 ^a^	7.22 ± 0.95 ^a^
	F5	4.56 ± 0.12 ^a^	0.44 ± 0.01 ^a^	3.17 ± 0.0029 ^a^	7.28 ± 0.80 ^a^
	F1	4.62 ± 0.16 ^a^	0.42 ± 0.07 ^a^	3.83 ± 0.29 ^b^	9.18 ± 0.02 ^b^
	F2	4.65 ± 0.07 ^a^	0.42 ± 0.07 ^a^	3.33 ± 0.29 ^ab^	7.89 ± 1.0 ^a,b^
3	F3	4.65 ± 0.13 ^a^	0.41 ± 0.001 ^a^	3.00 ± 0.003 ^a^	7.34 ± 0.41 ^a^
	F4	4.60 ± 0.06 ^a^	0.40 ± 0.01 ^a^	3.00 ± 0.003 ^a^	7.52 ± 0.93 ^a^
	F5	4.62 ± 0.10 ^a^	0.42 ± 0.06 ^a^	3.50 ± 0.005 ^ab^	8.37 ± 1.4 ^a,b^
	F1	4.92 ± 0.05 ^a^	0.35 ± 0.02 ^a^	3.83 ± 0.76 ^a^	10.95 ± 0.09 ^b^
	F2	4.83 ± 0.23 ^a^	0.42 ± 0.07 ^a^	3.83 ± 0.29 ^a^	9.11 ± 0.89 ^a,b^
6	F3	4.69 ± 0.10 ^a^	0.42 ± 0.01 ^a^	3.50 ± 0.005 ^a^	8.29 ± 1.5 ^a^
	F4	4.69 ± 0.14 ^a^	0.42 ± 0.04 ^a^	3.70 ± 0.003 ^a^	8.87 ± 1.2 ^a,b^
	F5	4.85 ± 0.09 ^a^	0.42 ± 0.06 ^a^	4.17 ± 0.0029 ^a^	10.02 ± 0.28 ^a,b^
	F1	4.88 ± 0.23 ^a^	0.36 ± 0.01 ^a^	4.33 ± 0.29 ^a^	11.99 ± 0.04 ^b^
	F2	4.89 ± 0.20 ^a^	0.40 ± 0.01 ^a^	4.17 ± 0.29 ^a^	10.34 ± 1.1 ^a,b^
9	F3	4.80 ± 0.15 ^a^	0.38 ± 0.08 ^a^	4.00 ± 0.006 ^a^	10.46 ± 1.7 ^a,b^
	F4	4.83 ± 0.08 ^a^	0.42 ± 0.001 ^a^	4.00 ± 0.005 ^a^	9.57 ± 1.0 ^a^
	F5	4.96 ± 0.10 ^a^	0.41 ± 0.05 ^a^	4.33 ± 0.0029 ^a^	10.48 ± 1.26 ^a,b^
	F1	5.07 ± 0.06 ^b^	0.38 ± 0.02 ^a^	4.83 ± 0.29 ^a^	12.65 ± 0.03 ^b^
	F2	5.09 ± 0.06 ^b^	0.39 ± 0.03 ^a^	4.67 ± 0.29 ^a^	12.00 ± 0.57 ^a,b^
12	F3	5.00 ± 0.05 ^b^	0.36 ± 0.12 ^a^	4.00 ± 0.003 ^a^	11.04 ± 0.96 ^a,b^
	F4	4.94 ± 0.08 ^a^	0.39 ± 0.07 ^a^	4.00 ± 0.008 ^a^	10.21 ± 2.0 ^a^
	F5	4.99 ± 0.09 ^ab^	0.40 ± 0.08 ^a^	4.67 ± 0.0029 ^a^	11.81 ± 0.92 ^ab^

**Table 3 polymers-12-01822-t003:** Evolution of weight loss percentage evolution in tomatoes with CS+RGEO treatments: F1 = control, F2 = CS, F3 = CS+ RGEO 0.5%, F4 = CS+RGEO 1.0%, and F5 = CS+RGEO 1.5%.

Day	0	3	6	9	12
Formulation					
F1	0	13.5 ± 0.2 ^b^	21.7 ± 0.2 ^c^	24.3 ± 0.3 ^c^	29.8 ± 0.2 ^c^
F2	0	8.6 ± 0.4 ^a^	11.4 ± 0.1 ^a,b^	20.0 ± 0.5 ^c^	20.0 ± 0.3 ^b^
F3	0	8.3 ± 0.3 ^a^	11.7 ± 0.2 ^b^	13.3 ± 0.4 ^a^	16.7 ± 0.4 ^a^
F4	0	9.1 ± 0.2 ^a^	10.9 ± 0.2 ^a^	13.6 ± 0.5 ^a^	18.2 ± 0.2 ^a,b^
F5	0	8.3 ± 0.4 ^a^	10.0 ± 0.3 ^a^	12.5 ± 0.2 ^a^	16.7 ± 0.1 ^a^

Mean values and intervals of LSD 95% according to the ANOVA test. Note: The superscript letter in the same column (a–c) refers to the significant differences (*p* < 0.05) between each treatment.

**Table 4 polymers-12-01822-t004:** The color coordinates L (lightness), *a* * (−green, +red), *b* *(−blue, +yellow), the red − yellow ratio (*a* */*b* *) and color difference score (Δ*E*) for tomatoes during the 12 days of storage with the CS + RGEO treatments: F1 = control, F2 = CS, F3 = CS + RGEO 0.5%, F4 = CS + RGEO 1.0%, and F5 = CS + RGEO 1.5%. The superscript letter (a–e) refers to the significant differences (*p* < 0.05) of LSD 95% according to the ANOVA test between each treatment in the same column.

Time	Treatment	L	*a* *	*b* *	*a* */*b* *	Δ*E*
0	**F1**	52.33 ^a^	18.83 ^b^	27.83 ^e^	0.68	
	**F2**	49.00 ^a^	22.33 ^d^	21.50 ^c^	1.04	
	**F3**	48.83 ^a^	17.50 ^a^	25.00 ^d^	0.70	
	**F4**	46.83 ^a^	20.50 ^c^	19.33 ^b^	1.06	
	**F5**	46.50 ^a^	22.33 ^d^	22.66 ^a^	0.99	
3	**F1**	48.66 ^b^	31.66 ^c^	42.33 ^e^	0.75	19.71
	**F2**	44.66 ^a,b^	33.00 ^d^	37.50 ^c^	0.88	19.72
	**F3**	43.83 ^a,b^	29.00 ^a^	39.50 ^d^	0.73	19.17
	**F4**	42.16 ^a,b^	31.5 ^b^	37.00 ^a^	0.85	21.33
	**F5**	41.00 ^a^	34.16 ^e^	39.83 ^b^	0.86	21.56
6	**F1**	45.50 ^a^	39.50 ^b^	59.16 ^e^	0.67	38.15
	**F2**	40.50 ^a^	44.66 ^d^	58.33 ^c^	0.77	43.90
	**F3**	41.33 ^a^	39.33 ^a^	58.50 ^d^	0.67	40.68
	**F4**	41.66 ^a^	44.83 ^e^	57.00 ^a^	0.79	45.14
	**F5**	40.66 ^a^	43.00 ^b^	61.00 ^b^	0.70	43.95
9	**F1**	27.5 ^a^	42.16 ^c^	56.66 ^d^	0.74	44.63
	**F2**	22.16 ^a^	44.33 ^d^	48.83 ^b^	0.91	44.17
	**F3**	24.83 ^a^	35.33 ^b^	56.0 ^c^	0.63	43.07
	**F4**	25.33 ^a^	45.50 ^e^	56.83 ^a^	0.80	49.93
	**F5**	25.33 ^a^	34.16 ^a^	51.33 ^e^	0.67	37.55
12	**F1**	18.00 ^a^	46.16 ^d^	58.66 ^d^	0.79	53.63
	**F2**	17.16 ^a^	49.50 ^e^	59.66 ^e^	0.83	56.64
	**F3**	17.33 ^a^	37.16 ^a^	54.16 ^c^	0.69	47.21
	**F4**	17.00 ^a^	43.00 ^c^	52.16 ^a^	0.82	49.74
	**F5**	17.00 ^a^	40.50 ^b^	55.50 ^b^	0.73	47.74

**Table 5 polymers-12-01822-t005:** Evolution of the Firmness (N) in tomatoes with CS + RGEO treatments: F1 = control, F2 = CS, F3 = CS + RGEO 0.5%, F4 = CS + RGEO 1.0%, and F5 = CS + RGEO 1.5%. Mean values and intervals of LSD 95% according to the ANOVA test.

Day	0	3	6	9	12
Formulation					
F1	108.2 ± 1.8 ^a^	10.0 ± 1.6 ^a^	2.5 ± 0.7 ^a^	1.6 ± 0.9 ^a^	1.6 ± 2.3 ^a^
F2	110.0 ± 4.4 ^a^	27.3 ± 3.5 ^ab^	22.4 ± 6.7 ^b^	14.4 ± 5.5 ^a,b^	10.3 ± 3.5 ^a^
F3	108.7 ± 2.1 ^a^	40.9 ± 6.5 ^b^	41.2 ± 4.6 ^c^	19.9 ± 1.8 ^b^	15.7 ± 4.2 ^a^
F4	110.2 ± 3.2 ^a^	69.5 ± 9.0 ^c^	60.0 ± 3.0 ^d^	52.5 ± 5.8 ^c^	39.2 ± 3.2 ^b^
F5	115.1 ± 5.5 ^a^	112.6 ± 7.2 ^d^	83.5 ± 1.2 ^e^	61.0 ± 4.9 ^c^	43.8 ± 5.1 ^b^

Mean values and intervals of LSD 95% according to the ANOVA test. Note: The superscript letter in the same column (a–e) refers to the significant differences (*p* < 0.05) between each treatment.

**Table 6 polymers-12-01822-t006:** Effect of treatments on the concentration of aerobic mesophilic and counting of molds in tomatoes with CS + RGEO treatments: F1 = control, F2 = CS, F3 = CS + RGEO 0.5%, F4 = CS + RGEO 1.0%, and F5 = CS + RGEO 1.5%. Mean values and intervals of LSD 95% according to the ANOVA test.

Day	0	3	6	9	12
	**Mesophilic bacteria (log UFC/g)**				
F1	3.14 ± 0.15 ^c^	4.86 ± 0.10 ^c^	5.80 ± 0.13 ^c^	6.01 ± 0.22 ^c^	6.96 ± 0.44 ^d^
F2	2.70 ± 0.18 ^c^	4.60 ± 0.16 ^c^	5.22 ± 0.13 ^c^	5.65 ± 0.45 ^c^	6.73 ± 0.25 ^d^
F3	1.34 ± 0.12 ^b^	2.99 ± 0.14 ^b^	3.63 ± 0.29 ^b^	4.03 ± 0.15 ^b^	4.33 ± 0.10 ^c^
F4	N.D	1.71 ± 0.21 ^a^	1.86 ± 0.18 ^a^	2.34 ± 0.18 ^a^	2.88 ± 0.09 ^b^
F5	N.D	1.34 ± 0.12 ^a^	1.49 ± 0.08 ^a^	1.54 ± 0.16 ^a^	1.73 ± 0.10 ^a^
	**Molds (Log UFC/g)**				
F1	2.73 ± 0.12 ^c^	3.25 ± 0.08 ^c^	4.61 ± 0.19 ^c^	5.52 ± 0.06 ^d^	5.66 ± 0.04 ^d^
F2	1.48 ± 0.21 ^b^	2.51 ± 0.16 ^b^	3.48 ± 0.27 ^b^	4.06 ± 0.18 ^c^	5.31 ± 0.10 ^c^
F3	N.D	N.D	N.D	2.53 ± 0.14 ^b^	3.77 ± 0.10 ^b^
F4	N.D	N.D	N.D	N.D	N.D
F5	N.D	N.D	N.D	N.D	N.D

Mean values and intervals of Tukey’s 95% according to the ANOVA test. N. D No detected. Note: The superscript letter in the same column (a–d) refers to the significant differences (*p* < 0.05) between each treatment.

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
