# Peer review of "Reduction of Postharvest Quality Loss and Microbiological Decay of Tomato “Chonto” (Solanum lycopersicum L.) Using Chitosan-E Essential Oil-Based Edible Coatings under Low-Temperature Storage"

_polymers, 2020, doi:10.3390/polym12081822_

Round 1

Reviewer 1 Report

The authors should add description on emerging research on edible coatings and updates references, with particular regards to Chitosan-E Essential Oil-based 5 edible coatings. 

A graphical scheme of preparation of edible coatings should be added. 

Lines 221-222: introductive lines should be enlarged. 

Data in Table 4 should be better discussed. 

More update references should be added along the manuscript. 

Author Response

Comments and Suggestions for Authors

The authors should add a description of emerging research on edible coatings and updates references, with particular regards to Chitosan-E Essential Oil-based 5 edible coatings. 

R// We appreciate the reviewer's comments. A description of chitosan-essential oil nanoemulsions, advantages, and food applications was included (lines 75-89): 

In the recent years, chitosan-based nanoemulsions have emerged as an alternative to the conventional biofilms [28], these present some advantages such as to allow a higher transfer area and higher reaction rates, a higher solubility, improved bioavailability,  optical transparency. Besides, they can limit the non-essential reactions with other components in the case of the food applications, as well as inhibit degradation during and after consumption [29]. Different studies present the chitosan- essential oils based nanoemulsions as an alternative to avoid the decay of fruits, with the critical advantage not to generate changes in the organoleptic conditions of the foods where they are applied [28].

Some studies have reported the effect of chitosan-based nanoemulsions incorporated with nutmeg seed essential oils [30] and Zatariamuti flora essential oil [31] in strawberry, with thyme essential oil in avocado [32], and with lemongrass essential oil in grape berries [33]. In general, the emulsions presented good antimicrobial activity and physicochemical properties preservation such as color, firmness, total soluble solids, and weight in the fruits where they were applied. Regarding tomatoes, Robledo et al. [34] reported a decrease in the Botrytis cinerea growth in cherry tomatoes with the use of chitosan-thymol essential oil-based nanoemulsion as the coating.

A graphical scheme of preparation of edible coatings should be added. 

R// The graphical scheme was included in Figure 1. Preparation of film-forming emulsions, (a) A schematic view of the different steps emulsions preparation process, (b) a detailed picture of the emulsion preparation setup.

Lines 221-222: introductive lines should be enlarged. 

R// The introductive lines were enlarged between lines 245-247:

The physicochemical properties determine the quality, shelf life, sensory aspects, and the post-harvest handling of the fruits and are very important when evaluating the effectiveness of the coatings. The pH, titratable acidity, soluble solids, and mature index of the tomato fruits were determined at days 0t, 3rd, 6th, 9th, and 12th of storage at a cold temperature (4°C) (Table 2).

Data in Table 4 should be better discussed. 

R// We appreciate the reviewer's comment. Data in Table 4 was complemented with the inclusion of the L* coordinate and DE. Also, an in-depth discussion between lines 477-493 was included:

Also, fruits coated with CS+RGEO presented a less color change, with an increase in the b*, and a* coordinates. The increase could be associated with the fact that lycopene (related to the red color) and β-carotene (compared with the orange color) achieve their concentrations peaks in the full ripening [75]. With red color increasing in tomatoes, a decrease in the L* value was also observed, indicating the darkening of the red color. The intensification generally occurs during the ripening of the tomatoes, as is shown with the results of DE. It is evident from the matrix of color differences between F1 with F3, F4, F5, the color change above 5.39 to the control can be perceived by consumers and associated with a higher ripening stage than coated tomatoes [76]. The chromophore degradation molecules like lycopene could be the main reason for the loss in color attributes, which could be delayed by coatings [27,33]. Metabolic reactions allow the color of the fruit to increase its intensity after chlorophyll degradation and lycopene synthesis [30]. From the results of DE, it is evident that coating has a beneficial effect on the reduction of color changes in tomatoes color change above 5.39 which was found in the control batch can be perceived by consumers and could be associated with a higher ripening stage of the tomatoes [84]."

More updated references should be added along with the manuscript. 

R// The references were updated.

Reviewer 2 Report

The reviewed manuscript includes discussion on a significant number of analyses performed on tomatoes coated with chitosan-based films. Taking into account the fact that most of the results are related to the changes in tomatoes, rather than polymers, I suggest publishing the manuscript in a different journal, for example, devoted to food processing.

Apart from the general conclusion, certain inaccuracies should be addressed and explained:

The maturity index methodology should be included as well as symbols used in equation number 4 should be explained.

The changes in the colour of tomatoes should be analyzed using the Delta E (ΔE) value. Describing the changes in particular parameters such as A and B is insufficient.

I suggest a publication of this particular manuscript in a different journal connected with the analysis of food.

Author Response

Comments and Suggestions for Authors

The reviewed manuscript includes a discussion on a significant number of analyses performed on tomatoes coated with chitosan-based films. Taking into account the fact that most of the results are related to the changes in tomatoes, rather than polymers, I suggest publishing the manuscript in a different journal, for example, devoted to food processing.

Apart from the general conclusion, certain inaccuracies should be addressed and explained:

The maturity index methodology should be included, as well as symbols used in equation number 4 should be explained.

R// We appreciate the reviewer's comment. The methodology used to determine the maturity index was included (Lines 165-169):

It was calculated using Equation (4) [17]:

1)

  (4)

Where, % BRIX is the total soluble solids (%) measured as degree Brix determined as is showed in the section 2.5.1 and %ACID is the titratable acid measured as citric acid % calculated in the section 2.5.2

The changes in the color of tomatoes should be analyzed using the Delta E (ΔE) value. Describing the changes in particular parameters such as A and B is insufficient.

R// We appreciate the reviewer's comment. The coordinate L and the Delta E (ΔE) value were included and discussed (lines 183-184; 297-303; 477-485).

3.2.7. Color parameters analysis

The analysis of the color in fruits impacts the marketing of the fruits. A consumer might be attracted or reject some fruit only by visual examination. We determined the color changes using the CIE L*a* b* scale, where a* and b* are the chromaticity coordinates (rectangular coordinates), and L* the lightness. On the other hand, +a* is the red direction, −a* is the green direction, +b* is the yellow direction, and –b* is the blue direction, and lightness value L* represents the darkest black at L*=0 and the brightest white at L* = 100. Table 4 shows a significant increase (p < 0.05) for the red/green coordinates (a*) and the yellow/blue coordinates (b*) until day 6th of the storage at low temperatures for all treatments. However, decreased values for the coordinate's degradation of the molecules responsible for the red color in tomatoes. Concerning the coordinate L*was observed a reduction with the storage time, and only it is presented significant differences among treatments F1 and F5 with F2, F3, and F4 for day 9th. At the end of the storage time on day 12th, the procedures with the lower values for coordinates a*, b*, and L* were F3, F4, and F5, indicating a slower ripening process. The control fruit and chitosan coating fruit showed high ΔE values, whereas, in treated fruits, this parameter was less affected. However, the best red/yellow ratios were exhibited by F2 and F4 and the minor DE by F3, F4, and F5, meaning that CS+RGEO did not negatively affect the color of tomatoes.

Also, fruits coated with CS+RGEO presented a less color change, with an increase in the b*, and a* coordinates. The increase could be associated with the fact that lycopene (related to the red color) and β-carotene (compared with the orange color) achieve their concentrations peaks in the full ripening [75]. With red color increasing in tomatoes, a decrease in the L* value was also observed, indicating the darkening of the red color. The intensification generally occurs during the ripening of the tomatoes, as is shown with the results of DE. It is evident from the matrix of color differences between F1 with F3, F4, F5, the color change above 5.39 to the control can be perceived by consumers and associated with a higher ripening stage than coated tomatoes [76]. The chromophore degradation molecules like lycopene could be the main reason for the loss in color attributes, which could be delayed by coatings [27,33]. Metabolic reactions allow the color of the fruit to increase its intensity after chlorophyll degradation and lycopene synthesis [30]. From the results of DE, it is evident that coating has a beneficial effect on the reduction of color changes in tomatoes color change above 5.39 which was found in the control batch can be perceived by consumers and could be associated with a higher ripening stage of the tomatoes [84]."

 I suggest a publication of this particular manuscript in a different journal connected with the analysis of food.

R// We appreciate the reviewer's suggestion. However, we answered to the call of the following special issue: Functional polymer coatings edited by Professor Ilker Bayer with a submission deadline May 31st, 2020. We though the topic follows well the special issue because of the application of the CS+RGEO coatings applied on tomatoes surfaces and monitoring the effect on the properties of the fruits and stability.
